# Pilates Method Improves Cardiorespiratory Fitness: A Systematic Review and Meta-Analysis

**DOI:** 10.3390/jcm8111761

**Published:** 2019-10-23

**Authors:** Rubén Fernández-Rodríguez, Celia Álvarez-Bueno, Asunción Ferri-Morales, Ana I. Torres-Costoso, Iván Cavero-Redondo, Vicente Martínez-Vizcaíno

**Affiliations:** 1Movi-Fitness S.L, Universidad de Castilla La-Mancha, 16002 Cuenca, Spain; ruben.fernandez12@alu.uclm.es; 2Health and Social Care Center, Universidad de Castilla La-Mancha, 16002 Cuenca, Spain; Ivan.Cavero@uclm.es (I.C.-R.); Vicente.Martinez@uclm.es (V.M.-V.); 3Universidad Politécnica y Artística del Paraguay, Asunción 001518, Paraguay; 4Faculty of Physiotherapy and Nursing, Universidad de Castilla-La Mancha, 45071 Toledo, Spain; Asuncion.Ferri@uclm.es (A.F.-M.); AnaIsabel.Torres@uclm.es (A.I.T.-C.); 5Facultad de Ciencias de la Salud, Universidad Autónoma de Chile, Talca 3460000, Chile

**Keywords:** aerobic capacity, cardiac rehabilitation, mind–body, Pilates, cardiorespiratory fitness, VO_2_ max, adults, prescription of exercise, systematic review, meta-analysis

## Abstract

Cardiorespiratory fitness has been postulated as an independent predictor of several chronic diseases. We aimed to estimate the effect of Pilates on improving cardiorespiratory fitness and to explore whether this effect could be modified by a participant’s health condition or by baseline VO_2_ max levels. We searched databases from inception to September 2019. Data were pooled using a random effects model. The Cochrane risk of bias (RoB 2.0) tool and the Quality Assessment Tool for Quantitative Studies were performed. The primary outcome was cardiorespiratory fitness measured by VO_2_ max. The search identified 527 potential studies of which 10 studies were included in the systematic review and 9 in the meta-analysis. The meta-analysis showed that Pilates increased VO_2_ max, with an effect size (ES) = 0.57 (95% CI: 0.15–1; I^2^ = 63.5%, *p* = 0.018) for the Pilates group vs. the control and ES = 0.51 (95% CI: 0.26–0.76; I^2^ = 67%, *p* = 0.002) for Pilates pre-post effect. The estimates of the pooled ES were similar in both sensitivity and subgroup analyses; however, random-effects meta-regressions based on baseline VO_2_ max were significant. Pilates improves cardiorespiratory fitness regardless of the population’s health status. Therefore, it may be an efficacious alternative for both the healthy population and patients suffering from specific disorders to achieve evidenced-based results from cardiorespiratory and neuromotor exercises.

## 1. Introduction

Strong evidence supports that higher levels of cardiorespiratory fitness (CRF) are associated with a lower risk of cardiovascular morbidity and mortality as well as all-cause mortality [1,2,3]. In addition, CRF decreases the risk of developing some specific diseases [4], such as chronic obstructive pulmonary disease (COPD) and lung or colorectal cancer [5,6], most of which are associated with a large burden of disease [7]. Furthermore, several studies have shown that higher levels of CRF may attenuate the negative association between CV risk factors and sedentary behaviours independent of physical activity [8,9,10,11]. Thus, CRF emerges as an independent predictor for several chronic diseases [12] and as a remarkable overall health status measure in different populations [12].

To improve CRF, current evidence suggests that physical exercise must reach a minimum intensity [13,14] of at least 45% oxygen uptake reserve in the general population and 70%–80% in athletes [15]. Greater improvements in maximal oxygen uptake (VO_2_ max) are obtained with vigorous physical exercises when compared with moderate intensity exercises [3]. Moreover, it has been suggested that some types of physical exercises that are not traditionally considered as cardiorespiratory exercises [16,17], such as Pilates, could increase CRF.

Pilates has become popular in recent years as a holistic exercise [16] focused on respiration, body control and accuracy of movements. Current evidence suggests positive effects of Pilates on respiratory muscle strength, balance, quality of life and overall physical performance [18,19,20,21,22,23,24]. These benefits are observed not only in the healthy population but also in those with specific disorders, such as chronic low back pain [16], multiple sclerosis [25], breast cancer [26] and Parkinson’s disease [27]. The neuromuscular stimulation achieved during Pilates [28] may be of sufficient intensity to improve CRF, providing benefits in VO_2_ max for individuals with different health conditions [29,30,31,32,33]. Thus, it seems that Pilates exercises include a mind–body component [34] that could have a beneficial impact in different populations.

However, evidence for the comparative benefits of Pilates vs. other physical exercises in terms of VO_2_ max remains inconclusive [22,35], and there are no studies that have evaluated oxygen consumption during Pilates sessions. Therefore, it is difficult to assess whether Pilates exercises reach the minimum intensity needed to improve CRF. We conducted this systematic review and meta-analysis to determine the effectiveness of Pilates on CRF as measured through VO_2_ max. Moreover, we explored whether the effect of Pilates on CRF could be modified by the participant’s health condition or baseline VO_2_ max level.

## 2. Materials and Methods

### 2.1. Search Strategy and Study Selection

The present review and meta-analysis were reported according to the Preferred Reporting Items for Systematic Reviews and Meta-Analyses (PRISMA) [36] and follow the recommendations of the Cochrane Handbook for Systematic Reviews of Interventions [37]. This study was registered through PROSPERO with registration number CRD42019124054. 

We conducted a systematic literature search in the following databases: MEDLINE (via PubMed), Cochrane Central Register of Controlled Trials (CENTRAL), EMBASE (via Scopus), Web of Science and the Physiotherapy Evidence Database (PEDro), from each database’s inception until September 2019 for studies aimed at determining the effectiveness of the Pilates method on CRF as measured through VO_2_ max. The search algorithm was conducted using PICO’s strategy (type of studies, participants, interventions, comparators and outcome assessment) and combined Medical Subject Headings, free-terms and matching synonyms of the following related words: (1) population: adults, “middle aged”, “young adult”; (2) intervention: Pilates, mind–body, “exercise movement techniques”; (3) outcome: “cardiorespiratory fitness”, “aerobic fitness”, “aerobic capacity”, “heart rate”; and (4) comparator: control conditions or another physical exercise. In addition, we searched the citations included in the identified publications deemed eligible for our study. The complete search strategy for MEDLINE is presented in Table 1.

### 2.2. Eligibility Criteria

Two initial reviewers (RFR and CAB) independently examined the titles and abstracts of retrieved articles to identify suitable studies. Those studies in which the title and abstract were related to the aim of the present review were included for full text request. We included studies that (1) were conducted as randomised controlled trials (RCTs), non-randomised controlled trials (non-RCTs) or pre-post studies; (2) included a mean participant age ≥18 years; (3) involved participants in any health condition; and (4) were based on at least one exercise intervention described as “Pilates” (mat, machine or both). Studies were excluded if (1) outcome measurements were not reported as VO_2_ max values, or (2) they were not written in English, Spanish or Portuguese. A third reviewer (VMV) resolved cases of initial reviewer disagreement.

#### Ethical Aspects

The present systematic review and meta-analysis were performed by collecting and analysing data from previous studies in which informed consent had been obtained by the respective original investigators. Therefore, this study was exempt from ethics approval.

### 2.3. Data Extraction and Quality Assessment

Two authors (RFR and CAB) independently extracted the following information from the included studies: First author’s name and year of publication; study design; characteristics of the participants included; mean age; sample size and percentage of female subjects; weekly frequency, period and modality of Pilates intervention; supervision of the intervention by a certified instructor; use of a detailed exercise protocol; the reported measurement of VO_2_ max; the device used to measure VO_2_ max; and main results. A third reviewer (VMV) resolved cases of author disagreement.

The risk of bias of RCTs was assessed using the Cochrane risk-of-bias tool for randomised trials (RoB 2.0) [38], in which five domains were evaluated: Randomization process, deviations from intended interventions, missing outcome data, measurement of the outcome, and selection of the reported result. Each domain was assessed for risk of bias. Studies were graded as (1) “low risk of bias” when a low risk of bias was determined for all domains; (2) “some concerns” if at least one domain was assessed as raising some concerns, but not to be at high risk of bias for any single domain; or (3) “high risk of bias” when high risk of bias was reached for at least one domain or the study judgement included some concerns in multiple domains [38].

For pre-post studies and non-RCTs we used the Quality Assessment Tool for Quantitative Studies [39], in which seven domains were evaluated: Selection bias, study design, confounders, blinding, data collection methods, withdrawals and dropouts. Each domain was considered strong, moderate or weak. Studies were classified as “low risk of bias” if they presented no weak ratings; “moderate risk of bias” when there was at least one weak rating; or “high risk of bias” if there were two or more weak ratings [39].

Risk of bias was independently assessed by two reviewers (RFR and CAB). A third reviewer (VMV) was consulted in case of disagreement.

### 2.4. Data Analysis

Primary data extracted from each study included mean VO_2_ max, standard deviation of pre-post intervention and sample size. Effect sizes (ES) and related 95% confidence intervals (CIs) were calculated for each study [40]. The Dersimonian and Laird random effects method [41] was used to compute pooled ES estimates and respective 95% CIs. We estimated the pooled ES for the effect of Pilates vs. the control group (CG). The heterogeneity of results across studies was evaluated using the I^2^ statistic, with I^2^ values of 0%–30% considered “not important” heterogeneity; >30%–50% representing moderate heterogeneity; >50%–80% representing substantial heterogeneity, and >80%–100% representing considerable heterogeneity. The corresponding *p*-values and 95%CI for I^2^ were also considered [42]. Finally, we conducted two additional analyses: (i) the pre-post ES of Pilates on the intervention group (Appendix A), and (ii) the mean difference of Pilates vs. CG (Appendix B).

For all the analyses, when studies reported data on two intervention groups of Pilates, the effects of both groups were pooled in order to calculate the average effect size. Finally, when studies reported more than one intervention, we only considered the Pilates intervention for conducting this meta-analysis.

A sensitivity analysis was conducted by removing each included study to assess the robustness of the summary estimates. Further, subgroup analysis based on participants’ health status and random-effects meta-regression by baseline VO_2_ max values were conducted to determine their potential effect on the pooled ES estimates. Finally, publication bias was evaluated through visual inspection of funnel plots and Egger’s regression asymmetry test for the assessment of small-study effects [43]. Statistical analyses were performed using StataSE software, version 15 (StataCorp, College Station, TX, USA).

## 3. Results

### 3.1. Systematic Review

#### 3.1.1. Study Selection

The search strategy identified 527 potential studies for inclusion. Of these, 10 studies were included in the systematic review. Only nine studies were included in the meta-analysis because one study [44] did not provide the required data to calculate ES (Figure 1).

#### 3.1.2. Study and Intervention Characteristics

Study and intervention characteristics are summarised in Table 2. Of the 10 included studies, five were RCTs [22,29,33,35,45], two were non-RCTs [31,44] and three were pre-post studies [30,32,46]. All the studies were conducted between 2008 and 2019 and included a total of 332 participants, of which 223 were in Pilates groups (67%) and 109 in control groups (33%). The age of the participants ranged between 18 and 66 years; four studies were conducted in women only [22,31,32,46]. Furthermore, seven studies were conducted in a healthy population, including people described in the primary studies as people without health disorders or specific pathologies [22,31] (four in sedentary individuals [30,32,44,46] and one in trained runners [33]) and three studies were conducted in populations with specific health disorders, including those described in the primary studies as suffering some diseases or specific health disorders such as heart failure [35], chronic stroke [29] and overweight/obesity [45].

In control groups, participants were encouraged to continue with their routine physical activity or to obtain conventional treatment. Among control groups, two studies did not allow structured physical exercise [22,45]; one did not describe the control group activity [31]; and one performed the running conventional program [33] and two studies the conventional rehabilitation programs [29,35].

Concerning the characteristics of the Pilates interventions, the majority of studies consisted of two or three 40–60 min sessions, three times per week, over 8–16 weeks. The mean attendance at the Pilates sessions was 88.2% (80%–100%). Among the 10 studies, six described the Pilates intervention as Pilates mat [22,29,30,33,35,46], three studies combined both modalities (mat and machine) [32,44,45] and one did not report the Pilates modality [31]. Moreover, six studies were conducted by a certified instructor [22,29,30,33,35,45] or with a detailed exercise protocol [29,30,32,34,44,45].

The outcome, VO_2_ max, was directly measured in nine studies (two with a cycloergometer and seven with a treadmill) [29,30,31,32,33,35,45,46,47] and one study [22] used an algorithm based on heart rate to estimate VO_2_ max values. The studies assessed participants at the end of the Pilates intervention, an no study measured VO_2_ max during the Pilates session.

#### 3.1.3. Quality Assessment and Risk of Bias

Five RCTs were assessed according to the RoB 2.0 tool [38], of which two were assessed as “low risk of bias” and three as “some concerns” (Figure 2). The remaining five studies (non-RCTs and pre-post studies) were assessed according to the Quality Assessment Tool for Quantitative Studies [39], of which two were classified as “low risk of bias”, two as “moderate risk of bias” and one as high risk of bias (Figure 3).

### 3.2. Data Synthesis

#### 3.2.1. Meta-Analysis

The pooled ES for the effect of Pilates vs. CG on CRF was 0.57 (95% CI: 0.15–1.00; I^2^ = 63.5%, *p* = 0.02) (Figure 4) and for Pilates pre-post ES was 0.51 (95% CI: 0.26–0.76; I^2^ = 67%, *p* < 0.01) (Figure A1, Appendix A). The mean difference analysis of Pilates vs. CG was 2.77 (95% CI: 1.12–4.42; I^2^ = 33.4%, *p* = 0.19) (Figure A3, Appendix B).

#### 3.2.2. Sensitivity and Meta-Regression Analyses

After removing studies from the analyses individually, none substantially modified the pooled ES estimate in Pilates vs. CG (Table 3), Pilates pre-post effect on intervention (Table A1, Appendix A) and mean difference of Pilates vs. CG. (Table A5, Appendix B). The subgroup analyses by participants’ health conditions modified the pooled ES estimate for Pilates vs. CG (Table 4) and mean difference of Pilates vs. CG (Table A6, Appendix B), but not for Pilates pre-post effect on intervention (Table A2, Appendix A).

The random-effects meta-regression models by VO_2_ max baseline levels were significant for Pilates vs. CG (*p* = 0.03) (Table 5) and for Pilates pre-post effect on intervention (*p* = 0.05) (Table A3, Appendix A) but not for mean difference of Pilates vs. CG (*p* = 0.08) (Table A7, Appendix B).

#### 3.2.3. Publication Bias

A significant publication bias was not found in Pilates vs. CG studies, as evidenced by both the funnel plot (Figure 5) asymmetry and an Egger’s test (*p* = 0.465) (Table 6), nor in the mean difference of Pilates vs. CG by funnel plot asymmetry (Figure A4, Appendix B) and an Egger’s test (*p* = 0.69) (Table A8, Appendix B). However, in Pilates pre-post effect studies publication bias was found (*p* = 0.07) (Table A4, Appendix A).

## 4. Discussion

This systematic review and meta-analysis were performed to determine the effectiveness of Pilates interventions for improvement of CRF measured through VO_2_ max. Our findings highlight that Pilates is an alternative exercise to improve VO_2_ max values. Furthermore, our results were substantially modified by participants’ health conditions for Pilates vs. control group analyses but not for Pilates pre-post effect on intervention; otherwise, baseline VO_2_ max values could influence CRF improvement.

Although some studies [22,35,45,46] have failed to show significant changes in CRF after Pilates intervention, no study has reported negative effects of Pilates on the CRF levels, and therefore the positive clinical implications should not be underestimated. Additionally, more significant benefits of Pilates on CRF were achieved when other activities, such as running, were included [33] and this could be explained through a synergistic relationship between these training methods.

Evidence suggests that people with lower levels of CRF are more sensitive to improvement of this parameter [47]. Accordingly, in our study estimates of pooled ES were higher in those studies in which participants had lower baseline CRF levels, such as people with health disorders. Conversely, our meta-regression analyses suggested that higher levels of VO_2_ max at baseline are related with higher ES of the Pilates intervention. These findings should cautiously be interpreted since they may indicate that the effect of Pilates in those studies with higher VO_2_ max levels at baseline were distortedly overestimated. Probably these biased estimates were a consequence of reporting results in absolute terms (change in VO_2_ max in ml) instead of in relative terms (percentage of increase in VO_2_ max), but could have clinical implications suggesting that Pilates exercise is an effective rehabilitation strategy for several disorders, including some cardiac pathologies. Moreover, Pilates exercise showed high compliance levels indicating that it may be better tolerated than the aerobic exercises typically employed in rehabilitation programs.

Three potential sources of improvement may explain the positive impact of Pilates intervention on CRF: Strengthening of the lumbopelvic region, increased flexibility of the ribcage and breathing exercises. First, the strengthening of lumbopelvic and core muscles induced by Pilates may produce a more efficient movement pattern in upper and lower limbs, as well as greater strength in expiratory muscles [19,33]. Second, due to the flexibility improvement, a more efficient mobility pattern of the ribcage may be achieved [30]. Finally, the breathing techniques adopted during Pilates training may increase lung capacity [29] and functionality of intercostal muscles [17]. On these bases, improved ventilation efficiency would be achieved, resulting in a higher flow of oxygenated blood into muscle tissues [35], enhanced local circulation [19,30] and muscle oxidative capacity [45], and less energy waste. Therefore, Pilates could reach the minimum intensity required to improve CRF [13,14] although no published study has verified this.

Our systematic review and meta-analysis present some limitations that must be stated. First, it was not possible to blind Pilates interventions and some of the included studies did not provide details about the randomisation sequence or allocation concealment. Second, considerable levels of heterogeneity were observed in the analyses, and we cannot omit this fact. Third, the heterogeneity of participants’ health conditions and the dose and intensity of the Pilates intervention could potentially affect our estimates. Fourth, significant publication bias was evidenced by Egger’s test and unpublished results could modify the findings of the present meta-analysis. Fifth, it should be highlighted the difficulty to comply with a full training program by very busy professionals, thus, this concern should not be neglected in the implementation of our results. Sixth, rarely it is possible to measure VO_2_ max directly in clinical settings, thus other more applicable procedures for indirect measurement of VO_2_ max should be used. Seventh, although subgroup analyses by participants’ health conditions modified the ES estimates, these results should be cautiously considered due to the lack of studies in each subgroup. Finally, due to the lack of long-term assessments, we could not determine whether the benefits to CRF measured through VO_2_ max are preserved over time. Therefore, our results should be cautiously considered.

## 5. Conclusions

In summary, our results support Pilates as an effective intervention to improve CRF in both healthy people and individuals with disorders related to aerobic capacity. Despite this, further studies should be conducted, including short- and long-term measurements to determine the intensity level reached by VO_2_ max during Pilates intervention and whether CRF improvement is preserved over time.

## Figures and Tables

**Figure 1 jcm-08-01761-f001:**
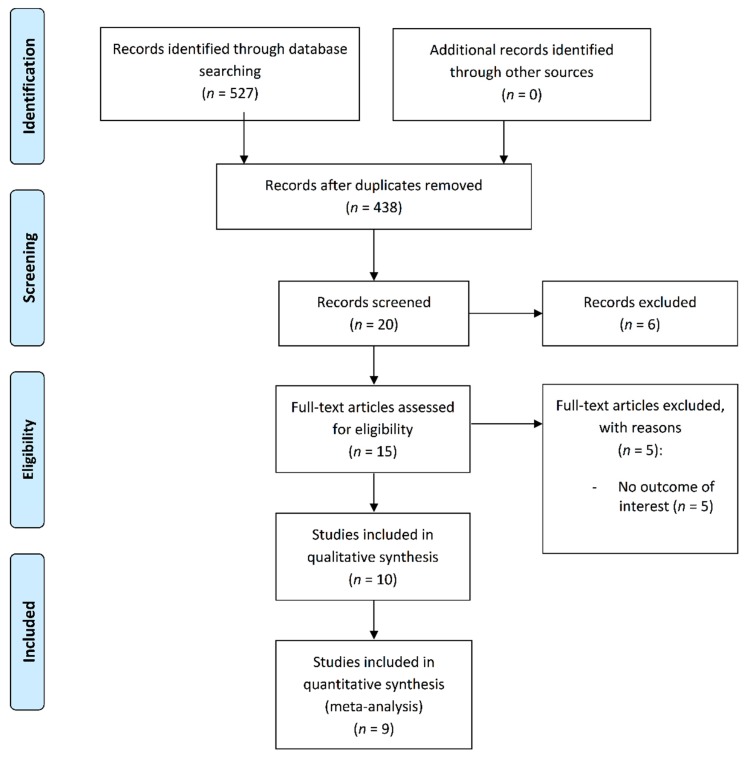
Flow of the included studies.

**Figure 2 jcm-08-01761-f002:**
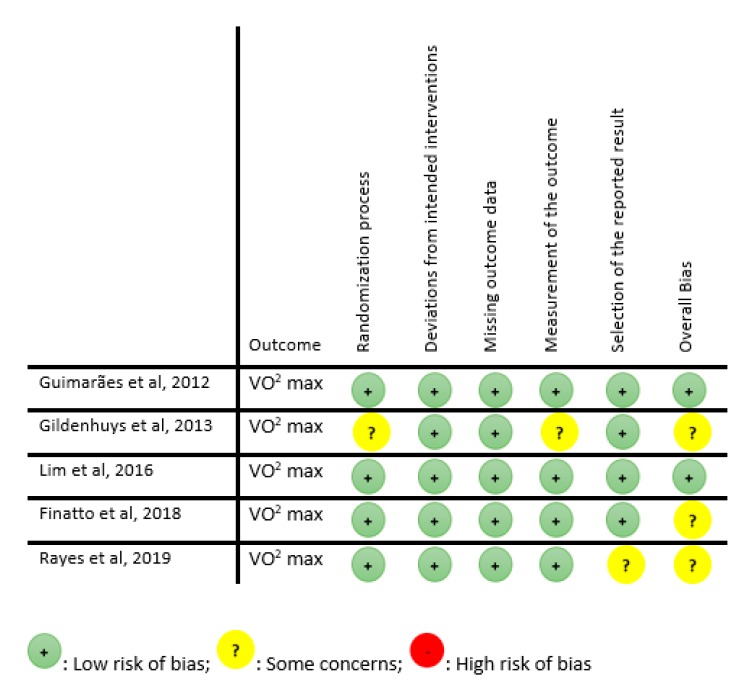
Quality assessment for RCT (RoB 2.0).

**Figure 3 jcm-08-01761-f003:**
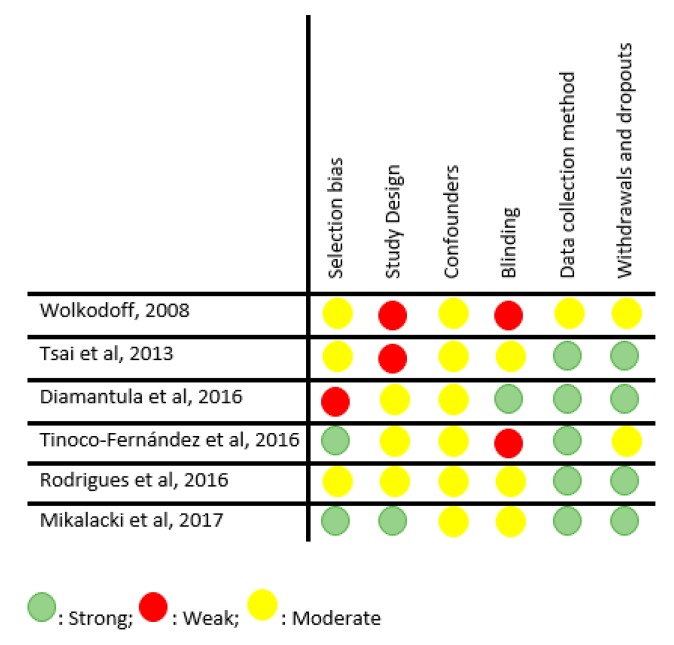
Quality assessment for non-RCT.

**Figure 4 jcm-08-01761-f004:**
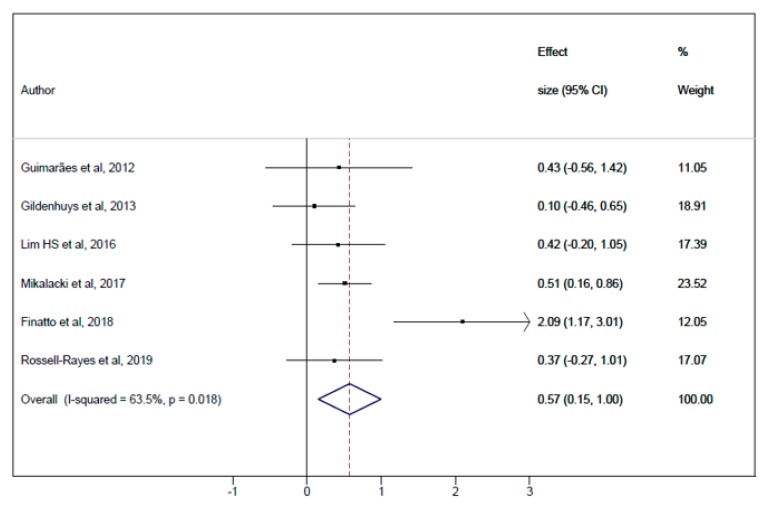
Meta-analysis for Pilates Method vs. control group (pooled ES analysis).

**Figure 5 jcm-08-01761-f005:**
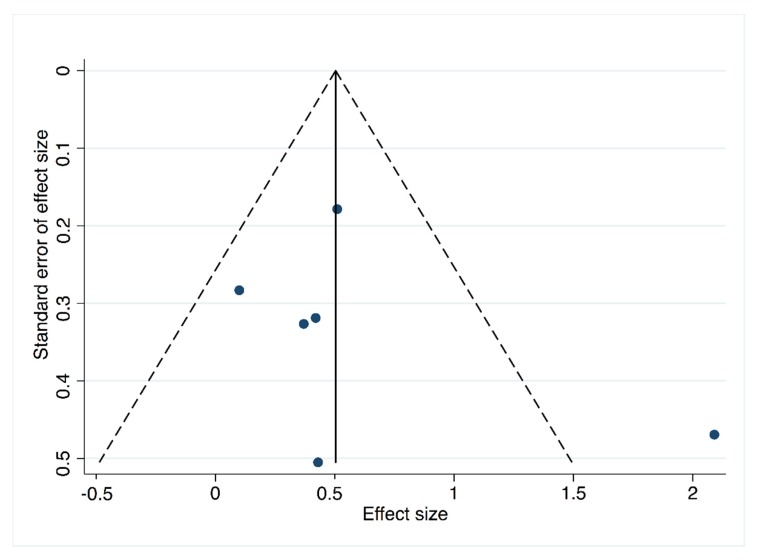
Funnel plot for Pilates vs control group.

**Table 1 jcm-08-01761-t001:** Strategy for MEDLINE.

Population	Intervention	Outcome
Adults	Pilates	“Cardiorespiratory fitness”
OR	OR	OR
Middle aged	Mind-body	“Aerobic fitness”
OR	OR	OR
Young adult	Exercise Movement Techniques (Mesh)	“Aerobic capacity”
		OR
		“Heart rate”
		OR
		Cardiorespiratory fitness (Mesh)

**Table 2 jcm-08-01761-t002:** The included studies.

Author	Design	Participants’ Characteristics	Mean Age	Sample Size (% Female)	Frequency	Period	Type of Pilates	Certified Instructor	Detailed Protocol	Outcome Measure	Outcome Results
Wolkodoff 2008 [44]	CT	Sedentary (healthy)	PG = 23–64	*n* = 20 PG = 14 (85.7%) CG = 6 (83.3%)	40′/3.2xwk	8wks	Both	NA	Yes	-Peak VO_2_ mL/kg/min (Oxycon Mobile)	CG change = 0.38 PG change = 6.06 17% of change in PG
Guimarães et al., 2012 [35]	RCPT	Heart failure	PG = 46 ± 12 CRG = 44 ± 11	*n* = 16 PG = 8 (38%) CRG = 8 (19%)	60′/2xwk	16wks	Mat	Yes	Yes	-Peak VO_2_ mLO_2_/kg/min (Vmax 229 model, SensorMedics, Yorba Linda, CA, USA)	PG: improvements in peak VO_2_ (*p* = 0.01) Comparing both groups, PG showed greater improvement on peak VO_2_ (*p* = 0.02)
Gildenhuys et al., 2013 [22]	RCT	Elderly women (healthy)	PG = 66 ± 5 CG = 65 ± 5	*n* = 50 PG = 25 (100%) CG = 25 (100%)	60′/3xwk	8wks	Mat	Yes	NA	-VO_2_ max mL.kg^−1^ min^−1^ (6minWalk; indirect equation)	PG did not significantly improve VO_2_ max (*p* = 0.247)
Lim HS et al., 2016 [29]	RCT	Chronic stroke	PG = 63 ± 8 CG = 62 ± 7	*n* = 20 PG = 10 (40%) CG = 10 (50%)	3xwk	8wks	Mat	Yes	Yes	-VO_2_ max mL/min -VO_2_ max per kg (metabolic analyzer: Quark b2, COSMED, Italy 2011)	PG: VO_2_ max and VO_2_ max per kg increased significantly CG: VO_2_ max per kg diminished significantly
Diamantoula et al., 2016 [46]	Q-E	Sedentary women (healthy)	PG = 26 ± 5 AP = 21.3 ± 2	PG land = 20 (100%) AP = 20 (100%)	2xwk	2years	Mat/aqua	NA	NA	-VO_2_ max mL/min (Ergometer cycle (Amila kh803), following the Astrand-Ryhming test, based on heart rate in submaximal effort)	No differences between groups, better VO_2_ max in total for both groups
Tinoco- Fernández et al., 2016 [30]	Q-E	Sedentary students (healthy)	PG = 18–35	*n* = 45 PG = 45 (78%)	60′/3xwk	10wks	Mat	Yes	Yes	-VO_2_ max L/kg/min -VO_2_ max L/min (MasterScreen CPX apparatus)	Increment in peak VO_2_ and VO_2_ max
Rodrigues et al., 2016 [32]	Q-E	Sedentary women (healthy)	PG = 23 ± 2	PG = 10 (100%)	45′/2xwk	11wks	Both	NA	Yes	-VO_2_ max mL.kg^−1^ min^−1^ portable metabolic system (VO2000^®^, MedGraphics^®^, St. Paul, MN, USA)	Peak VO2 tended to increase, but the differences were not statistically significant
Mikalacki et al., 2017 [31]	CT	Adult women (healthy)	PG = 48 ± 7 CG = 47 ± 7	*n* = 64 PG = 36 (100%) CG = 28 (100%)	55–60′/2xwk	NA	NA	NA	NA	-Relative VO_2_ max -Absolute VO_2_ max (Medisoft, model 870c)	PG: significant increase on relative VO2max, absolute VO_2_ max -CG: not significant changes
Finatto et al., 2018 [33]	RCT	Trained runners (healthy)	PG = 18 ± 1 CG = 18 ± 1	*n* = 32 PG = 15–13 NA % CG = 16–15	60′/1xwk	12wks	Mat	Yes	NA	-VO_2_ max mL.kg^−1^.min^−1^ (VO2000 (Medgraphics, Ann Arbor, USA)	PG: significantly higher values on VO_2_ max (*p* < 0.001)
Rayes et al., 2019 [45]	RCT	Overweight/obese	PG = 55.9 ± 6.6 CG = 45.5 ± 9.3	*n* = 60 NA% PG = 22 CG = 25/17	60′/3xwk	8wks	Both	Yes	Yes	-VO_2_ max (mL/kg/min) (motorized treadmill; Inbrasport, ATL, Porto Alegre, Brazil)	PG: Significant improvement on VO_2_ max CG: not significant changes

CT: controlled trial; RCT: randomised controlled trial; RCPT: randomised controlled pilot trial; Q-E: quasi-experimental; PG: Pilates group; CG: control group; AP: Aqua-Pilates group; NA: not available; wk: week; VO_2_ max: maximal oxygen uptake.

**Table 3 jcm-08-01761-t003:** Sensitivity analyses.

Pilates Method vs. Control Author, Year	ES	LL	UL	I^2^
Guimarães et al., 2012 [35]	0.6	0.12	1.08	70.8
Gildenhuys et al., 2013 [22]	0.69	0.20	1.18	64.4
Lim HS et al., 2016 [29]	0.62	0.10	1.14	70.7
Mikalacki et al., 2017 [31]	0.62	0.03	1.22	70.8
Finatto et al., 2018 [33]	0.4	0.16	0.64	0
Rossell-Rayes et al., 2019 [45]	0.63	0.12	1.15	70.4

ES: Effect size; LL: Lower limit; UL: Upper limit.

**Table 4 jcm-08-01761-t004:** Subgroup analyses by participants’ health status.

Pilates Method vs. Control				
	ES	LL	UL	I^2^
Healthy	0.80	−0.05	1.65	85
Unhealthy	0.40	−0.01	0.81	0

ES: Effect size; LL: Lower limit; UL: Upper limit.

**Table 5 jcm-08-01761-t005:** Meta-regression analyses by VO_2_ max baseline values.

	Coefficient	*p*
Pilates Method vs. control	0.04	0.03 *

VO_2_ max: Maximal oxygen uptake (mL/kg/min); * Significant at *p* ≤ 0.05.

**Table 6 jcm-08-01761-t006:** Publication bias by Egger’s test.

	Coefficient	*p*-Value
Pilates method vs. control group	1.64	0.47

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
