# Peer review of "Pilates Method Improves Cardiorespiratory Fitness: A Systematic Review and Meta-Analysis"

_jcm, 2019, doi:10.3390/jcm8111761_

Round 1

Reviewer 1 Report

Timely review conducted in a very appropriate manner.

Fluent and easy to read.

Appropriate discussion and conclusion.

My only comment is that very busy professionals may struggle to comply with a full training program.  

Author Response

Reviewer 1:

Specific comment: Timely review conducted in a very appropriate manner. Fluent and easy to read. Appropriate discussion and conclusion. My only comment is that very busy professionals may struggle to comply with a full training program.

Authors’ response: We really appreciate the reviewer´s words. Regarding the comment, we agree with the fact that very busy professionals may struggle to comply with a full training program, therefore we have recognised this as a limitation of the implementation of the present study and included a statement in limitation section:

 “Fifth, it should be highlighted the difficulty to comply with a full training program by very busy professionals, thus, this concern should not be neglected in the implementation of our results.”  

Reviewer 2 Report

The cardiorespiratory fitness is an important factor, that improve all cause mortality. The paper"Pilates method..." is systematic review and meta analysis. The methodology is very good, the results are comprehensive. I didn´t find the difference between "healthy" population and population with some diseases.. I will add it in the results and discusion.

Author Response

Reviewer 2:

Specific comment: The cardiorespiratory fitness is an important factor, that improve all-cause mortality. The paper "Pilates method..." is systematic review and meta-analysis. The methodology is very good, the results are comprehensive. I didn´t find the difference between "healthy" population and population with some diseases. I will add it in the results and discussion.

Authors’ response: We appreciate the reviewer’s comments and suggestions. We have added an explanation of the terms “healthy population” and “population with some diseases” in the results as follows: 

“… including people described in the primary studies as people without health disorders or specific pathologies [22, 31] (4 in sedentary individuals [30,32,44,46] and one in trained runners [33]) and 3 studies were conducted in populations with specific health disorders including those described in the primary studies as suffering some diseases or specific health disorders such as heart failure [35], chronic stroke [29] and overweight/obesity [45].”

Reviewer 3 Report

GENERAL CONSIDERATIONS

Thank you for giving me the opportunity to review this relevant paper.

The Authors conducted a meta-analysis to investigate the effectiveness of Pilates intervention in improving cardiorespiratory fitness through VO2max improvement.

The methodological value of the paper is high, but some points need careful considerations.

This meta-analysis has a double research question?

- The effect of Pilates on VO2max compared to the control group (unpaired)

- The effect of Pilates on VO2max in the same group of patients, pre and post intervention (paired)

While the first is a meta-analysis of intervention obviously, the second design is a pre-post meta-analysis.

The effect measure adopted is the effect size. VO2max is a continuous variable. Should it be reported as standardized mean difference (SMD)?

Can we use standardized mean difference for the second question? We need to know or to impute a correlation coefficient?

I have some doubts about this methodology.

To conduct this meta-analysis with a double question, we have to include also single-group design with pre and post intervention (Pilates) assessment.

In my opinion, this is difficult for readers and makes the “project more complicated.

Excluding the pre-post intervention should be an option. The authors should include studies with at least 2 groups (pilates vs control) and assess the effect as SMD, based on changes in pre-post scores. It is enough and allow a simple interpretation of the results.

Introduction

- “The neuromuscular stimulation achieved……..CRF”  This sentence is not supported by reference 28.

-  The sentence about ACSM is an opinion (considering also the previous incorrect sentence). Reference n.3 does not include Pilates and it couldn’t include it based on previous statement.

Eligibility criteria:

“mean participant age >18”. This means that you have included subject with < 18 years probably. It is correct?

How have you managed studies with more than 2 groups of treatments? i.e. Pilates vs control vs other treatment?

Study and intervention characteristics

- check the number of studies and their description. “Furthermore, 7 studies……overweight/obesity”

- which activities are allowed in control group? Physical exercise? Nothing? This topic is crucial in the present meta-analysis.

- Treadmill and cycloergometer use algorithms to assess VO2max also. It is almost impossible to measure VO2max directly in common clinical practice. Specify better this problem.

Sensitivity and meta-regression analyses

“The subgroup analyses by participants’ health conditions did not…….” This statement is not in line with table 4 results. The table include negative LL (-0.05 and -0.01) This means that LL and UL include null value 0, determining a not singnificative p value (not reported).

Table 5

This is the result of the meta-regression. Outcome variable: effect size. The regression is made about baseline VO2max. The “bias coefficient” should be the point estimate. Is it should be negative? The effect should decrease, increasing the pre VO2max at baseline.

Publication Bias

Funnel plot not shown.

Discussion

The discussion should be designed on the new results, after their revision. The discussion section should be more accurate and complete, including more references and a detailed discussion about the results of the present study and the available results in literature.

Author Response

Reviewer 3:

Specific Comment: This meta-analysis has a double research question? The effect of Pilates on VO2max compared to the control group (unpaired) The effect of Pilates on VO2max in the same group of patients, pre and post intervention (paired). While the first is a meta-analysis of intervention obviously, the second design is a pre-post meta-analysis. The effect measure adopted is the effect size. VO2max is a continuous variable. Should it be reported as standardized mean difference (SMD)? Can we use standardized mean difference for the second question? We need to know or to impute a correlation coefficient? I have some doubts about this methodology. To conduct this meta-analysis with a double question, we have to include also single-group design with pre and post intervention (Pilates) assessment. In my opinion, this is difficult for readers and makes the “project more complicated. Excluding the pre-post intervention should be an option. The authors should include studies with at least 2 groups (pilates vs control) and assess the effect as SMD, based on changes in pre-post scores. It is enough and allow a simple interpretation of the results.

Authors’ response: We really appreciate the reviewer’s comments and suggestions. Considering that the included studies reported cardiorespiratory fitness using different measurement techniques, we have calculated the pooled effect size and also, as suggested, we have included an analysis on the mean difference in supplementary material. Thus, the information of this analysis has been included in the manuscript as follows:  

“Finally, we conducted two additional analyses: i) the pre-post ES of Pilates on the intervention group (Appendix C), and ii) the mean difference of Pilates vs CG (Appendix D).”

Regarding the last comment, and in order to clarify the results section, we have removed the meta-analysis of pre-post effect intervention of the main paper and included it in the supplementary material to display the fullest picture of the available evidence. Consequently, we have modified methods, results and discussion sections.

Specific comment: Introduction. “The neuromuscular stimulation achieved during Pilates may be of sufficient intensity to improve CRF”. This sentence is not supported by reference 28.

Authors’ response: We apologise for the inconvenience. The wrong reference (number 28) has been removed.

Specific comment: The sentence about ACSM is an opinion (considering also the previous incorrect sentence). Reference n.3 does not include Pilates and it couldn’t include it based on previous statement.

Authors’ Response: Thank you for your comment. We have modified the sentence and included a new reference to support the following statement:

Thus, it seems that Pilates exercises include a mind-body component [34] that could have a beneficial impact in different population.” 

Specific comment: mean participant age >18”. This means that you have included subject with < 18 years probably. It is correct?

Authors’ response: Thank you for the comment. The ‘mean participant age > 18’ refers to the mean age of the participants included in the primary studies reported by their respective authors. Only one study (Finatto et al, 2018) reported a mean age of 18±1, but as they state in their inclusion criteria, participants were: male, practice of running for at least six months before the study, with experience in 5-km running races, age between 18 and 28 years, and absence of medical restrictions. Therefore, no study includes participant less than 18 years.

Specific comment: How have you managed studies with more than 2 groups of treatments? i.e. Pilates vs control vs other treatment?

Authors’ response: Thank you for the comment. As suggested, we have included information about how we managed those studies including more than 2 groups of treatment as follows:

For all the analyses, when studies reported data on two intervention groups of Pilates, the effects of both groups were pooled in order to calculate the average effect size. Finally, when studies reported more than one intervention, we only considered the Pilates intervention for conducting this meta-analysis.”

Specific comment: check the number of studies and their description. “Furthermore, 7 studies……overweight/obesity”

Authors’ response: Thank you for the comment. Done.

Specific comment: Which activities are allowed in control group? Physical exercise? Nothing? This topic is crucial in the present meta-analysis

Authors’ response: We would like to thank the reviewer’s comment. We have included information regarding the activities in the control groups in the results section.

“Among control groups, two studies did not allow structured physical exercise [22, 45]; one did not describe the control group activity [31], one performed the running conventional program [33] and two studies conventional rehabilitation programs [29,35].”

Specific comment: Treadmill and cycloergometer use algorithms to assess VO2max also. It is almost impossible to measure VO2max directly in common clinical practice. Specify better this problem.

Authors’ response: The reviewer’s comment seems thoughtful. We have addressed this concern in limitations section as follow.

Sixth, rarely it is possible to measure VO2 max directly in clinical settings, thus other more applicable procedures for indirect measurement of VO2 max should be used.”

Specific comment: “The subgroup analyses by participants’ health conditions did not…….” This statement is not in line with table 4 results. The table include negative LL (-0.05 and -0.01) This means that LL and UL include null value 0, determining a not significative p value (not reported).

Authors’ response: We would like to apologize for the mistake. We have properly modified the results section and include a statement in limitations section recommending caution in the interpretation of the subgroup analyses results due to the scarcity of studies.

“Seventh, although subgroup analyses by participants’ health conditions modified the ES estimates, these results should be cautiously considered due to the lack of studies in each subgroup.”

Specific comment: Table 5. This is the result of the meta-regression. Outcome variable: effect size. The regression is made about baseline VO2max. The “bias coefficient” should be the point estimate. Is it should be negative? The effect should decrease, increasing the pre VO2max at baseline.

Authors’ response: We really appreciate the comment of the reviewer. We have properly modified the discussion section and include the following statement:

Evidence suggests that people with lower levels of CRF are more sensitive to improvement of this parameter [47]. Accordingly, in our study estimates of pooled ES were higher in those studies in which participants had lower baseline CRF levels, such as people with health disorders. Conversely, our meta-regression analyses suggested that higher levels of VO2 max at baseline are related with higher ES of the Pilates intervention. These findings should cautiously be interpreted, since they may indicate that the effect of Pilates in those studies with higher VO2 max levels at baseline were distortedly overestimated. Probably these biased estimates were a consequence of reporting results in absolute terms (change in VO2 max in ml) instead of in relative terms (percentage of increase in VO2max)…”

Specific comment: Publication bias, funnel plot not shown.

Authors’ response: Thank you for the comment. We have added the funnel plot figure as supplementary material.

Specific comment: The discussion should be designed on the new results, after their revision. The discussion section should be more accurate and complete, including more references and a detailed discussion about the results of the present study and the available results in literature.

Authors’ response: The discussion section has been properly modified according the reviewers’ comments. Moreover, some suggested limitations have been included.

Round 2

Reviewer 3 Report

Dear authors,

thank you for your changes.

I think that the present form of the manuscript is more detailed than the previous. 

Although data indicate a limited ability of Pilates in improving CRF, the effect is significative.

Further studies will be able to confirm your results and to give them robustness, providing more details about protocol of Pilates, frequency and volume of intervention. 

Thank you